# The effect of tart cherry juice compared to a sports drink on cycling exercise performance, substrate metabolism, and recovery

**Ruirui Gao, Nicole Rapin, Justin W. Andrushko[ID], Jonathan P. Farthing, Julianne Gordon, Philip D. Chilibeck[ID]** *

College of Kinesiology, University of Saskatchewan, Saskatoon, Canada

* phil.chilibeck@usask.ca

**Data Availability Statement:** All relevant data are within the manuscript.

## Abstract

Tart cherries have low glycemic index, antioxidant and anti-inflammatory properties, and therefore may benefit performance and recovery from exercise. We determined the effects of consuming tart cherry juice versus a high-glycemic index sports drink on cycling performance, substrate oxidation, and recovery of low-frequency fatigue. Using a randomized, counter-balanced cross-over design, with one-month washout, 12 recreational cyclists (8 males and 4 females; 35±16y; $VO_{2peak}$ 38.2±7.4 ml/kg/min) consumed cherry juice or sports drink twice a day (300mL/d) for 4d before and 2d after exercise. On the exercise day, beverages (providing 1g/kg carbohydrate) were consumed 45min before 90min of cycling at 65% $VO_{2peak}$, followed by a 10km time trial. Blood glucose, lactate, carbohydrate and fat oxidation, respiratory exchange ratio (RER), $O_2$ cost of cycling, and rating of perceived exertion (RPE) were measured during the initial 90min of cycling. Muscle soreness, maximal voluntary contraction (MVC) and low-frequency fatigue were determined at baseline and after the time trial on the exercise day, and 30min after beverage consumption 24 and 48h later. There were no differences for time trial performance (17±3min cherry juice vs. 17±2min sports drink, $p = 0.27$) or any other measures between drink conditions. There were time main effects ($p<0.05$) for isometric MVC (decreasing) and low-frequency fatigue (increasing; i.e. decreased force at low relative to high stimulation frequencies), changing significantly from baseline to post-exercise and then returning to baseline at 24h post-exercise. Tart cherry juice was not effective for improving performance, substrate oxidation during exercise, and recovery from exercise, compared to a high-glycemic index sports drink.

## Introduction

Many studies have shown that tart cherry juice consumption can inhibit inflammation and enhance recovery after exercise [1, 2]; however, very few studies investigated the effects of tart cherries on exercise performance [3]. Tart cherry juice, compared with high glycemic index (GI) beverages, might enhance exercise performance via (1) the preservation of carbohydrate availability during exercise, especially during later stages of prolonged exercise, because of its

**Funding:** This study was funded by the Canadian Cherry Producers and Saskatchewan Academy of Sports Medicine Inc.

**Competing interests:** The authors have declared that no competing interests exist.

low GI; (2) its antioxidant capacity by reducing oxidative stress and rebalancing redox [1]; (3) and its potential for inducing vasodilation by increasing oxygen delivery to skeletal muscles via enhanced blood flow, which is manifested in reduced oxygen consumption at a given workload and reduced blood pressure [4].

Tart cherries are rich in anthocyanins and flavonoids that have a protective effect against inflammation and lipid peroxidation caused by intense exercise, therefore they may reduce exercise-induced muscle damage and enhance recovery of muscle function after exercise. However, studies on this topic assessed muscle damage by measuring muscle proteins in the blood, inflammation, and muscle function changes, which are indirect and inaccurate measures of muscle damage [1, 2]. A more direct measurement of muscle damage can be obtained by applying electrical stimulation at different frequencies to a muscle before and after intense exercise, and calculating a ratio of low frequency force to high frequency force output [5]. This study used this direct measurement (i.e., ratio of low frequency force output to high frequency force output) as a muscle damage indicator after cycling exercise. Although eccentric contractions are typically used in studies to induce muscle damage [1, 2], high-intensity concentric contractions induced by repetitive cycling still has the potential to induce muscle damage [6, 7].

The purpose of this study was to compare the effect of tart cherry juice consumption (GI = 45) [8] to a high glycemic index sports drink (GI = 89) [9] for improving cycling endurance performance and recovery from exercise in recreational cyclists. We also compared the beverages for their effect on post-exercise changes in blood pressure to see if enhancing blood flow with cherry juice might have health benefits. It was hypothesised that tart cherry juice, compared with high glycemic index sports drink, would result in: (1) a faster time trial performance; (2) a lower rise in blood glucose when consumed before exercise, and greater fat oxidation and lower carbohydrate oxidation during cycling exercise; (3) a better recovery from exercise as indicated by lower sensations of muscular soreness, higher force output during a test of isometric maximal voluntary contractions (MVC), and higher force output during low frequency stimulation immediately after, and 24 and 48 hours after a cycling test; (4) lower $O_2$ cost of exercise and (5) a lower blood pressure response in the 24 hours after exercise testing.

## Methods

### Participants

Twelve recreational cyclists (8 males, 4 females, 35±16 years of age, 69.5±10 kg, 171.2±6.2 cm, $VO_{2peak}$ 38.2±7.4 ml/kg/min) participated in this study. A sample size calculation based on the performance results from a previous study that was most similar to ours [4], with their standardized effect size (1.05), alpha of 0.05 and power of 80%, indicated that a sample size of 10 would be required for us to detect significant differences for cycling time trial performance. Inclusion criteria were: 18 years of age or older, and cycling for at least 30 minutes, three times per week. Exclusion criteria included allergy to cherries, history of major knee injuries, currently taking any medication or nutritional supplementations, and history of cardiovascular, renal, or gastrointestinal disease. This study was approved by the University of Saskatchewan Biomedical Research Ethics Board (reference number Bio#16–273) and was registered at clinicaltrials.gov (NCT03313388). The study was performed according to the ethical standards as laid down in the 1964 Declaration of Helsinki and its later amendments. Written consent was obtained from all individual participants included in the study. The recruitment period for this study was from May 2018 to July 2018.

## Study design

The study used a counterbalanced, double-blind, cross-over design where participants were randomized to a tart cherry juice or sports drink condition, and then participated in the other condition a month later. Participants and all personnel involved in outcome assessments were blinded to the beverage conditions. A researcher who had no role in data collection or analysis prepared the beverages so that blinding was achieved. Many human exercise studies have successfully found beneficial effects of tart cherry supplementation; however, little rational has been provided for the supplementation protocol, and an optimal strategy is not known. The loading phase in these studies range from 4 to 7 days before exercise, the day of exercise (usually approximately 2 hours before exercise), and 2 to 4 days after exercise [10]. Most of these studies used cyclists and tested cycling performance. Some studies assessed running performance when runners were involved. The dosage used in these studies were 237 to 355 mL of cherry juice (or 30 mL of concentrate) twice a day [10]. Therefore, the supplementation strategy used in this study was that participants consumed cherry juice or sports drink for 4 days before (twice a day, 300mL/d), on the day of (1g/kg carbohydrate provided by beverages; 45 min before exercise) and 2 days after (twice a day, 300mL/d) cycling exercise.

There were eight study visits. Visit #1 consisted of familiarization testing for muscle soreness, MVC of the knee extensor muscle group, determination of force output at low and high frequencies of electrical stimulation (i.e. 20 and 80 Hz) of the knee extensors, and an aerobic capacity test on a cycle ergometer (Monark Ergomedic 874E, Vansbro, Sweden) to determine fitness level and workload to be used for subsequent testing. The maximal workload and $VO_{2peak}$ were determined using a progressive cycling test as described elsewhere [11]. Muscle soreness was assessed with a method described by Barss et al. [12]: Using an Algometer (WAGNER Instruments, FDK 60 Force Dial; Greenwich, CT), 5kg of pressure was applied on four different sites of the knee extensors. While the participants were seated upright with a 90-degree bend in the knee, the distance between the anterior superior iliac crest and proximal edge of the patella was measured. Four locations were marked with a permanent marker: 25, 50, 75 and 100% of the distance between the two anatomical landmarks with 0% indicating the proximal landmark (anterior superior iliac crest). Participants reported pain on a 100mm visual analog scale, with 0mm indicating no pain and 100 mm indicating maximal pain. A total pain score was derived by taking an average of the pain from all four sites. Visit #1 also included a practice of a 10km time trial on a cycle ergometer, which was the performance indicator.

Visit #2 consisted of a familiarization trial using the actual cycling test used in the study on a cycle ergometer (Monark Ergomedic 874E, Vansbro, Sweden), (i.e. 90 minutes of cycling at an intensity corresponding to 50% of the maximal workload reached during the aerobic capacity test, corresponding to ~65%$VO_{2peak}$) immediately followed by a 10km time trial. During the 10 km time trial, the cycle display screen was masked so that participants were blinded to the cadence, speed and elapsed time. Participants were allowed to see the elapsed distance. The cycling test (i.e. 90 minutes at constant load + 10 km time trial) was designed to simulate the distance (i.e. 40 km) that a cyclist typically covers during an Olympic distance triathlon.

After visit #2, participants were randomized to receive either a tart cherry juice beverage (Everyday Farms—Aunt Mary's Genuine Tart Cherry Juice, Prairie Fruit Processors Ltd., Melfort, SK, Canada; Table 1) or sports drink (Gatorade Perform® Fruit Punch Thirst Quencher Powder, Pepsico, Canada; Table 1) using a computer-generated random number table. A small amount of food dye (i.e. 4 mL) was added to the sports drink to make the drink identical in colour to the cherry juice, along with a small amount (i.e. 4 mL) of calorie-free lemon concentrate so the drinks would have a similar taste. These were consumed twice a day (150 mL

**Table 1. Composition of two drinks.**

| | On days 1–4, 6 and 7 | | On day 5 for a 70kg individual | |
|---|---|---|---|---|
| | Tart cherry juice | Sports drink | Tart cherry juice | Sports drink |
| GI | 45 | 89 | 45 | 89 |
| Total volume (mL) | 150 | 150 | 503 | 503 |
| Calories (Kcal) | 101 | 100 | 339 | 280 |
| Carbohydrate (g) | 22 | 25 | 70 | 70 |
| Fiber (g) | 1 | 0 | 4 | 0 |
| Sugars (g) | 13 | 24 | 40 | 68 |
| Fat (g) | 1 | 0 | 2 | 0 |
| Protein (g) | 1 | 0 | 4 | 0 |
| Anthocyanins (mg) | 1380 | 0 | 4625 | 0 |

per dose at breakfast and before going to bed) for 4 days to deliver an appropriate dose of fla-vonoids (about 9.2 mg/ml anthocyanins) in the cherry juice condition to reduce inflammation and oxidation [13]. The amount of calories to be consumed in the sports drink condition was matched to the cherry juice condition on these days. (Insert Table 1 here)

Two days before visit #3 (i.e. on the 3rd and 4th day of the cherry juice/Gatorade supple-mentation) participants were instructed to try to minimize the amount of polyphenols they consumed in their diet by minimizing intake of fruits, vegetables, tea, coffee, alcohol, choco-late, cereals, wholemeal bread, and grains because these foods contain some of the same benefi-cial ingredients as the cherry juice. Participants kept a food diary and a physical activity log by recording all foods and drinks they consumed, and their physical activity for these two days. These diaries were photocopied and given back to participants before the opposite drink con-dition. Participants were asked to repeat the same diet and physical activity in the diaries for these two days in the next phase. They were asked to refrain from strenuous exercise (except for the cycling exercise in the present study) between the 3rd day of the beverage supplementa-tion and the last day of the measurement of muscle recovery for each of the two conditions. Participants were also asked not to change their eating habits and physical activity in any way between the 3rd day of the beverage supplementation and the last day of the measurement of muscle recovery for each of the two conditions. They were asked not to drastically change their physical activity habits throughout the entire study.

Visit #3 occurred on the 5th day of cherry juice/sports drink supplementation. Participants came into the lab after a 10 hour fast. They were tested again for muscle soreness, MVC, and force output at low and high frequencies of stimulation. They were then given 1.0 grams of car-bohydrates per kg body mass in the form of either cherry juice or Gatorade (Table 1) 45 min-utes before exercise testing. The timing and dose were to optimize carbohydrate availability prior to exercise [14] because anthocyanin (the main flavonoid in tart cherry juice) bioavail-ability increases to a peak between 1–2 h post-ingestion [15], a time which would coincide with the exercise testing. Blood glucose from a fingertip blood sample was assessed on a gluc-ometer (Accu-chek Compact Plus, Roche Diagnostics, Mannheim, Germany) before the drink and 10, 20, 30, and 40 minutes after the drink consumption. The exercise testing involved 90 minutes of cycling at 50% of the workload reached on the aerobic capacity test, followed by a 10 km time trial where participants covered 10 km as fast as they could. To ensure participants pedaled at the required power output for the duration of the 90 minutes, the researchers care-fully monitored participants' pedaling rate on the cycle ergometer throughout the 90 minutes. Recovery from a cycling test of this duration (i.e. approximately 105 minutes) benefits from tart cherry juice supplementation [13] and this duration of cycling is adequate to cause low-

frequency fatigue [16]. During the 90 minutes of cycling (before the time trial) blood glucose and lactate (Lactate Scout+, EKF Diagnostics, Leipzig, Germany) were assessed from fingertip blood samples at 15, 30, 60, and 90 minutes. Rating of perceived exertion (RPE) was assessed at the same time points. Respiratory gases were collected for 5-minute durations starting at 10, 25, 55, and 85 minutes of cycling for determination of respiratory exchange ratio (RER), expired carbon dioxide ($VCO_2$) and oxygen consumption ($VO_2$) with open-circuit indirect calorimetry (Vmax Encore 29, Viasys respiratory Care Inc., Palm Springs, California, USA), and for assessment of $O_2$ cost of cycling. Carbohydrate and fat oxidation were calculated using Brouwer Constants Equations [17]. After the time trial, muscle soreness, MVC, and force output at low and high stimulation frequencies were again determined. Participants were then fitted with a 24-hour blood pressure monitor (Oscar 2™system, SunTech Medical®, Inc., USA) which they wore overnight and into the next day. The blood pressure cuff was wrapped to the upper non-dominant arm. Based on participants' self-reported sleeping hours at night, the monitor was set to measure blood pressure every 30 minutes when participants were awake and every 2 hours during sleeping at night. Daytime blood pressure was calculated as the average blood pressure from 9:00am to 10:00pm whereas nighttime blood pressure was calculated as the average blood pressure from 12:00am to 6:00am.

Visit #4 to the lab occurred the next morning, again in a fasted state. The blood pressure cuff was removed. Thirty minutes after consumption of a 150 mL dose of the cherry juice or calorie/carbohydrate-matched sports drink, muscle pain, MVC, and force output at low and high frequencies were determined. Participants consumed 150 mL of the cherry juice or sports drink before bedtime. Visit #5 occurred the next morning and involved the same procedures as visit #4 (except that the blood pressure cuff was not worn during this last day).

Visits #6, 7, and 8 occurred a month later after 4 days of cherry juice or sports drink supplementation (i.e. the opposite drink condition to what the participant received prior to visit #3). All testing was identical to tests in visits #3, 4, and 5. A month between conditions was chosen to allow more than adequate wash-out of flavonoids from the cherry juice, to ensure that any females who participated in the study were doing the exercise testing at approximately the same phase of their menstrual cycle, and to minimize any potential repeated bout effect (i.e. where a bout of muscle-damaging exercise has a protective effect on a subsequent bout of exercise)[16].

## Neuromuscular fatigue

To assess neuromuscular fatigue, resting peak twitch torque, followed by MVC, and evoked torque from high (80Hz) and low (20Hz) frequency nerve stimulation was assessed in the right knee extensors. For each of these assessments participants were seated upright with 90 degrees of hip flexion and 80 degrees of knee flexion (0 degrees = maximal knee extension) on an isokinetic dynamometer (Humac NORM; CSMi, Stoughton, MA). Reproducibility of these measurements was as follows: MVC, intraclass correlation coefficient (ICC) range = 0.91–0.99, technical error of the measurement (TEM) = 9.7%; peak torque with 20 Hz stimulation (P20), ICC range = 0.80–0.98, TEM = 13.3%; peak torque with 80 Hz stimulation (P80), ICC range = 0.96–0.99, TEM = 6.3% [18].

Custom software in LabVIEW (version 8.6) was used to obtain stimulator pulses and torque for each of the nerve stimulation protocols (resting peak twitch torque and high- and low- frequency fatigue). A desktop computer equipped with an analog-to-digital converter was used to convert analog signals from each device to digital signals, which were displayed in LabVIEW and recorded to disk for later analysis.

**Electrical stimulation.** Electrically evoked contractions of the right knee extensors were achieved by stimulating the femoral nerve using a Constant Current High Voltage Stimulator

(model DS7AH, Digitimer, Hertfordshire, England) and 2 round circular electrodes (Ultra-Stim® X, USX2000, Axelgaard Manufacturing Co., Ltd., CA, USA) of 5cm diameter. The cathode electrode was placed on the anterior segment of the body, manually pressed into the femoral triangle by the experimenter while the anode was placed on the posterior segment of the body, in the gluteal fold.

**Resting peak twitch torque.** Supramaximal doublet stimulations (two 0.5 ms pulses, 80 Hz) were delivered to the femoral nerve to determine maximal twitch torque. A series of resting control twitches were used to determine the current (mA) required to reach a maximum resting doublet twitch torque. The procedure started with a very low level of current, barely detectable by the participant, and involved progressively raising the current (5–10 mA increments) until a plateau in twitch torque was observed, as detected by the dynamometer. This level of current, plus an additional 10%, was used to evoke maximal twitch torque. The current required to evoke a supramaximal resting twitch was reassessed during each visit. Three evoked twitches were recorded and then averaged to calculate peak twitch torque.

**Maximal isometric torque.** Following the assessment of resting peak twitch torque, three isometric MVCs of the knee extensors were completed on the isokinetic dynamometer. At each time point, participants performed three 3-second MVCs with a 1-minute rest period between attempts. Verbal encouragement was given during every trial and the contraction with the highest peak torque was used for further analysis.

**High- and low-frequency stimulation.** To assess peripheral fatigue, high- and low- frequency stimulation of the femoral nerve was used to evoke contractions of 0.5 seconds while the participant was instructed to completely relax their leg. High frequency stimulation (80 Hz; 40 stimuli, 0.5 ms pulse duration) preceded low frequency stimulation (20 Hz; 10 stimuli, 0.5 ms pulse duration). A submaximal current (70% of the supramaximal current delivered to acquire resting peak twitch torque) was used to deliver the high- and low- frequency stimuli. Submaximal current was used to minimize discomfort. Peak torque for each frequency (P80 and P20, respectively) and the P20/P80 torque ratio (low-frequency fatigue) were used for analysis.

## Statistical analysis

All statistical analyses were performed using SPSS version 21 (SPSS Inc., Chicago, DE, USA). Results are reported as mean ± standard deviation. Blood pressure and time trial results were assessed using paired *t*-tests. A two-factor repeated measures analyses of variance (ANOVA) was used to assess differences between conditions (tart cherry juice versus sports drink) and time (baseline, and 10, 20, 30, and 40 minutes) for blood glucose before exercise. A two-factor repeated measures ANOVA was used to assess differences between conditions (tart cherry juice versus sports drink) and time (15, 30, 60, and 90 minutes during cycling exercise) for blood glucose, lactate, and RPE. A two-factor repeated measures ANOVA was used to assess differences between conditions (tart cherry juice versus sports drink) and time (average of values from 10–14, 25–29, 55–59, and 85–89 minutes during the cycling exercise) for fat oxidation, carbohydrate oxidation. $VO_2$, and RER. A two-factor repeated measures ANOVA was used to assess differences between conditions (tart cherry juice versus sports drink) and time (pre-exercise, immediately post-exercise, and 24 and 48 hours post-exercise) for low frequency fatigue, MVC, and muscle soreness. We did not include sex as a factor in our analyses due to the small number of males and females in each group (i.e., 8 males and 4 females). Shapiro-Wilk tests were used to check the normality of data. Wilcoxon signed rank tests, as a nonparametric alternative for paired-t test, were used for overall systolic blood pressure and awake systolic blood pressure since there was a violation of normality. Data were log-transformed for

glucose during 90 min exercise, lactate, RPE, fat oxidation and MVC as there was a violation of normality for these variables. A post hoc test with Bonferroni correction was used to determine any differences between trials. The alpha level was accepted at $\alpha \leq 0.05$.

## Results

One male participant was unable to complete the time trial under the sports drink condition because they became ill during the time trial. One male was unwilling to wear the blood pressure monitor and two additional males and one female did not wear the monitor during one or both of the night times; therefore, these data were excluded from analyses. Two males had data that could not be analyzed from the low frequency fatigue assessment.

A significant condition × time interaction was found for glucose concentration before exercise, $F(4, 44) = 9.060$, $p < 0.001$ (Fig 1), with glucose concentration lower after cherry juice than sports drink consumption from 10 to 40 minutes post-consumption. This confirmed that tart cherry juice had a lower glycemic index than the sports drink.

All values are means (SD). GG, Greenhouse-Geisser; RPE, rating of perceived exertion; RER, respiratory exchange ratio. * Significantly different from tart cherry juice condition at the same time point, $p < 0.05/5$. N = 12 for all variables.

There were no differences for time trial performance (17±3min cherry juice vs. 17±2min sports drink, n = 11; $p = 0.27$), and blood pressure (Table 2) between tart cherry juice or sports drink conditions. There were no condition × time interactions for any of the variables measured before exercise, during exercise, or during recovery from exercise (Figs 1 and 2). There were time main effects ($p < 0.05$, Fig 1) for carbohydrate oxidation (decreasing over exercise, with values from 10–14 minutes and from 25–29 minutes greater than both 55–59 and 85–89 minutes), fat oxidation (increasing over exercise with values from 10–14 minutes lower than 55–59 and 85–89 minutes, and from 25–29 minutes lower than 85–89 minutes), RER (decreasing over exercise with values from 10–14 minutes greater compared to 55–59 and 85–89 minutes, and from 25–29 minutes greater compared to 85–89 minutes), RPE (increasing over exercise, although the Bonferroni-corrected post-hoc test showed no significant differences between individual time points), $O_2$ cost of exercise (i.e. $VO_2$) (decreasing over exercise with values from 10–14 minutes greater compared to 55–59 and 85–89 minutes and from 25–29 minutes greater compared to 85–89 minutes) and blood lactate (decreasing over exercise with the values at 15 minutes greater compared to 60 and 90 minutes, and the value at 30 minutes greater compared to 90 minutes). There were also time main effects ($p < 0.05$, Fig 2) for isometric MVC (decreasing) and low-frequency fatigue (n = 10; increasing; i.e. decreased force output at low relative to high stimulation frequencies) from before to after exercise and then recovering the next day.

A decrease in the P20: P80 Hz torque ratio indicates greater low-frequency fatigue. Values are means (SD). GG, Greenhouse-Geisser; MVC, maximal voluntary contraction. N = 12 for all variables except for low-frequency fatigue (n = 10).

## Discussion

The primary finding of this study is that low-glycemic index tart cherry juice consumption did not improve exercise performance, substrate metabolism or oxygen cost during exercise or recovery from exercise, compared to consuming a high-glycemic index sports drink. Performance during a 10 km time trial, RPE, carbohydrate and fat oxidation, RER, $VO_2$ at a given work rate, blood lactate accumulation, blood pressure, muscle soreness, MVC and low-frequency fatigue were not different between the two drink two conditions.

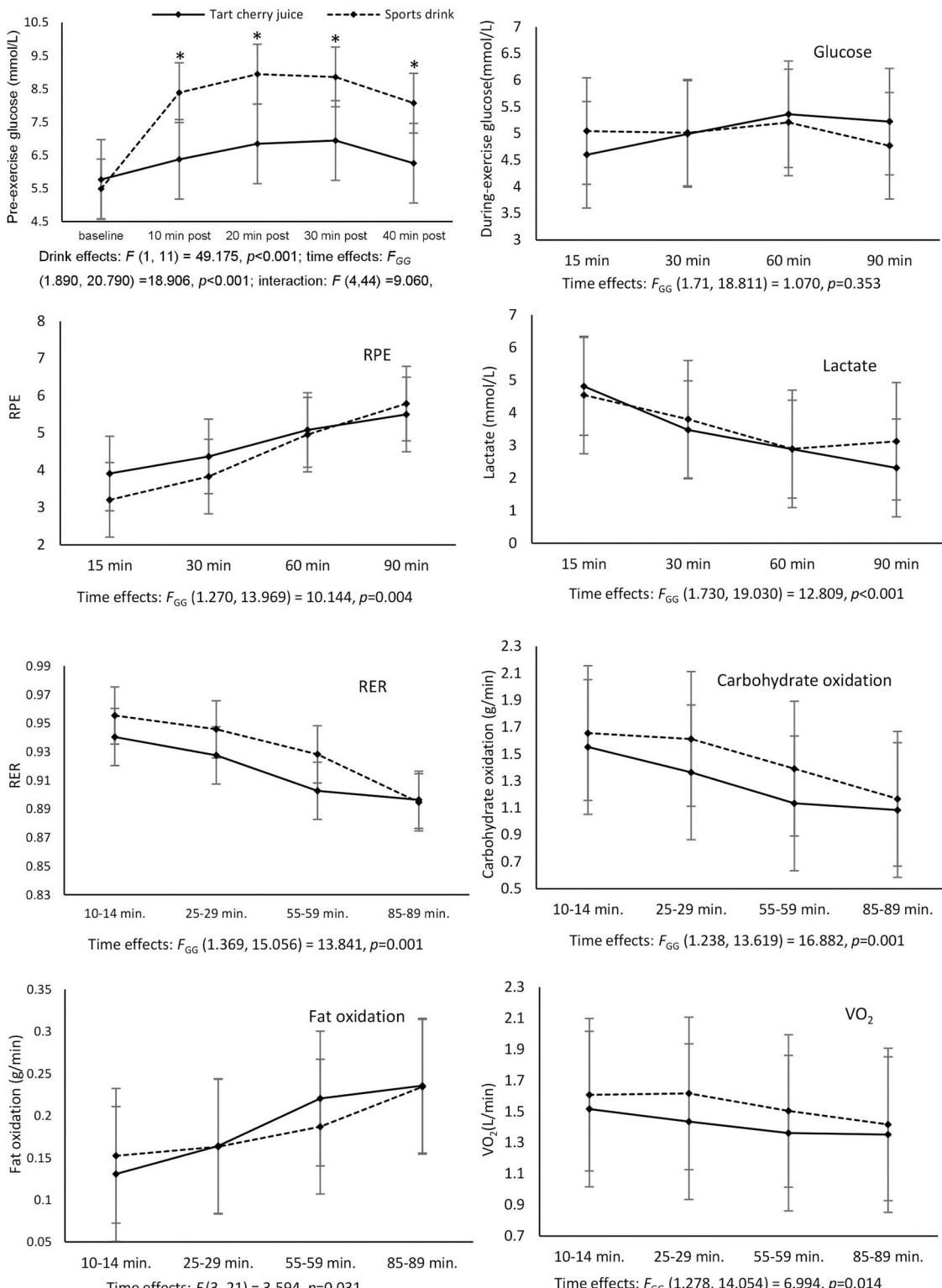

**Fig 1. Pre-exercise blood glucose after beverage consumption and changes in variables during exercise.** For glucose measurements, values on the x-axis represent "post-drink consumption".

Table 2. Blood pressure (BP) differences between drink conditions.

| | Tart Cherry Juice | Sports Drink | Statistical difference between conditions |
|---|---|---|---|
| Overall systolic BP (mmHg) (n = 11) | 118±8 | 118±7 | $p = 0.838$ |
| Overall diastolic BP (mmHg) (n = 11) | 68±6 | 67±6 | $p = 0.339$ |
| Awake systolic BP (mmHg) (n = 11) | 119±8 | 120±7 | $p = 0.838$ |
| Awake diastolic BP (mmHg) (n = 11) | 70±6 | 68±7 | $p = 0.341$ |
| Asleep systolic BP (mmHg) (n = 8) | 106±14 | 107±13 | $p = 0.651$ |
| Asleep diastolic BP (mmHg) (n = 8) | 58±9 | 57±8 | $p = 0.597$ |

Note: All values are means ± SD.

## Effects of tart cherry juice supplementation on exercise performance

We hypothesized tart cherry juice would improve exercise performance because tart cherry juice has a low glycemic index, antioxidant, and vasodilatory properties. The present study however did not find a benefit for performance, which might be due to several factors. Studies involving well-trained athletes with high $VO_{2peak}$ (e.g. $VO_{2peak}$ = 59-62mL/kg/min in three studies) found significant performance-enhancing effects of tart cherry juice [4, 13, 19], whereas in the present study, we assessed recreational cyclists with mean $VO_{2peak}$ of 38.2±7.4 ml/kg/min. Any supplementation-induced small effects might be better detected in elite athletes, compared with moderately trained athletes or recreational cyclists, since elite cyclists have a lower coefficient of variation for time trial performance than moderately trained cyclists [20, 21]. However, when we ran statistical analyses on the top six fittest participants, we still did not observe statistical differences for performance between conditions (cherry juice: 15.1 ± 1.0 min vs. sport drink 15.1 ± 2.1 min; $p = 0.99$). Our study indicates that perhaps the participants in our study were not limited by oxygen delivery because they were not trained athletes and therefore any vasodilatory effect of the cherry juice would not have an effect on performance. Previous studies assessing tart cherry juice assessed exercise performance with different measurements, e.g. cycling efficiency measured by oxygen consumption [13], half-marathon run (21.1 km) finish time [22], time to exhaustion during a severe-intensity cycling test and total work and peak power during a 60-seconds all-out sprint [19], and 15-km time trial [4]. There remains a debate surrounding the best measure of cycling performance as a large day-to-day variability exists when time to exhaustion is employed [23] whereas a recent study suggested time-to-exhaustion was better than time trials for measuring cycling performance [24].

## Effects of tart cherry juice supplementation on substrate metabolism

Theoretically, pre-exercise tart cherry juice consumption might increase fat oxidation and decrease carbohydrate oxidation because of its low glycemic index and high content of polyphenols. Low glycemic index foods or drinks induce a slower and smaller rise in insulin levels whereas after consumption of high glycemic index foods or drinks, insulin levels increase dramatically [14]. Insulin inhibits fat oxidation and promotes carbohydrate oxidation [25]. In the present study, however, tart cherry juice consumption had no differential effects on metabolism compared to a high-glycemic index sports drink. Cherries are high in fructose, which is specifically metabolized in the liver. High levels of fructose increase hepatic insulin resistance and may result in increased insulin levels [26]. A limitation of our study is that we did not

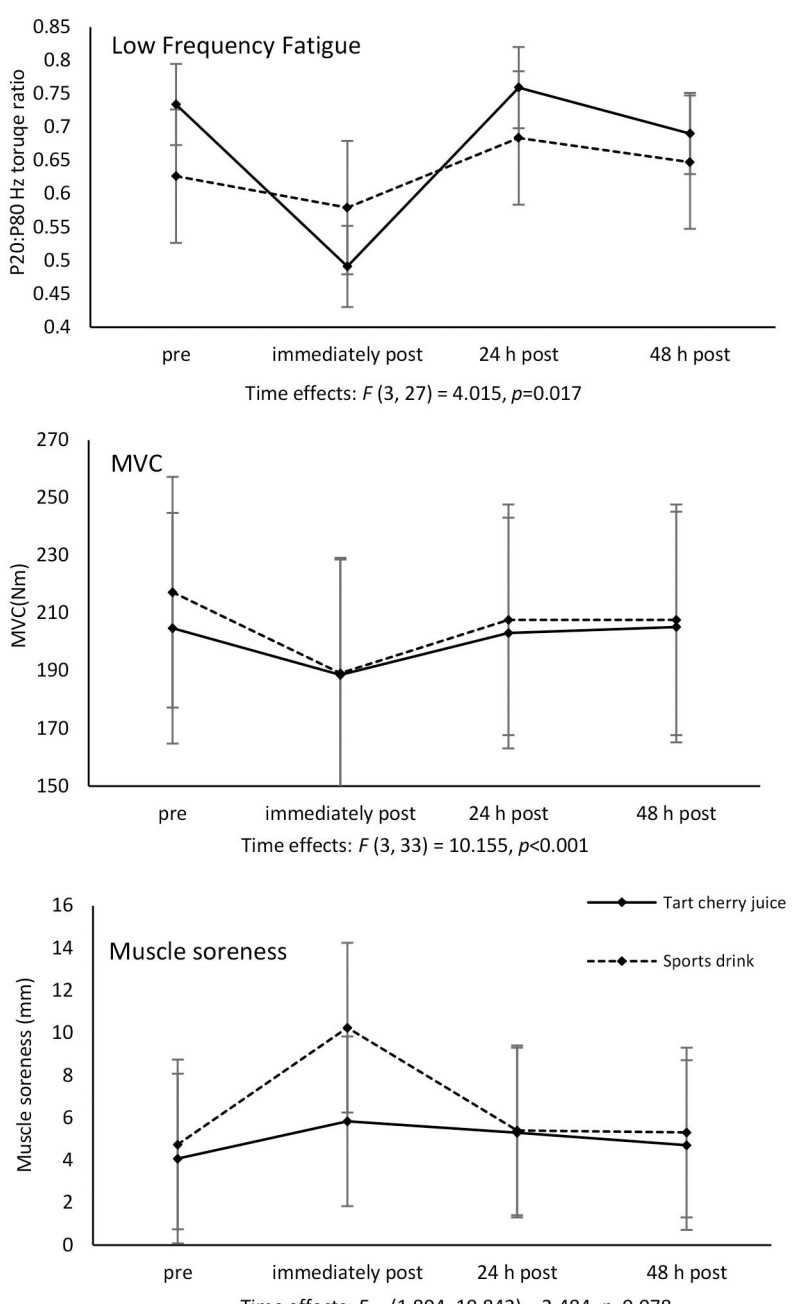

**Fig 2. Changes in low-frequency fatigue (P20:P80 Hz torque ratio), MVC and muscle soreness.**

assess insulin levels. Tart cherries are a rich source of polyphenols such as anthocyanins (mainly cyanidin-based anthocyanins) and catechin. Evidence that polyphenols might increase fat oxidation is provided by many studies using New Zealand blackcurrant [27, 28] and green tea extract [29, 30]. The improvement in fat oxidation during exercise induced by catechin polyphenols from green tea could be dose-dependent [29] and a lower dose of green tea extract did not increase fat oxidation during prolonged cycling [31]. Seven-day intake of anthocyanin-rich (mainly delphinidin-based anthocyanins) New Zealand blackcurrant extract showed a dose-dependent effect on improving fat oxidation during prolonged cycling exercise at 65%

$VO_{2max}$ in endurance-trained male cyclists [28]. The anthocyanin intake in the present study (2760 mg per day in cherry juice form) was higher than that used in the New Zealand blackcurrant extract studies (105 to 315 mg in capsule form) [27, 28]. However, an exact comparison of anthocyanin content in different studies requires caution since the possible variation in anthocyanin types (i.e. delphinidin-based anthocyanins and cyanidin-based anthocyanins), consumption form (i.e. capsule form and juice form), bioavailability and interactions with other nutrients contained in the supplementation may affect results. Therefore, the effects of tart cherry juice on substrate utilization might also be affected by dose; the dose used in the present study might not be high enough to alter substrate metabolism. The current study is limited in that we did not measure the plasma levels of anthocyanins or anthocyanins derived metabolites. Also, the level of anthocyanins in our cherry juice beverage was estimated based on the product label and not measured directly. There was a slight, but statistically significant decrease in $VO_2$ across our exercise test (across both drink conditions) (Fig 1). This is contrary to the $VO_2$ drift (i.e., slow component) that might be expected. The $VO_2$ slow component is correlated with lactate concentration [32]. Lactate concentration decreased throughout our exercise test indicating our participants were most likely working below their lactate threshold. We speculate that the reduced $VO_2$ observed during our exercise test may be related to the reduced lactate concentration, which is most likely due to lactate removal during exercise via the lactate shuttle or Cori Cycle [33].

Results are discrepant from studies investigating the effects of foods differing in glycemic index on lactate levels during exercise [14, 34]. In principle, the increased fat oxidation after low glycemic index food consumption might allow lower use of glycolysis, thus reducing lactate production [14]. Polyphenol antioxidants induce an increased vasodilation and blood flow, which increases oxygen delivery and clearance of lactate; therefore, tart cherry juice could theoretically induce a smaller rise in lactate levels during prolonged and intense exercise; however, no difference was found between conditions in the present study. This may also be explained by high content of fructose in cherry juice because fructose is converted into glucose and lactate in the liver [33, 35], thus potentially raising lactate levels. A limitation of our study is that we did not assess blood flow with more accurate measurement like a Doppler ultrasound. Instead, only blood pressure and lactate levels were measured.

### Effects of tart cherry juice supplementation on recovery from exercise

There were no differences between the two conditions for muscular soreness, MVC, or low-frequency fatigue immediately, and 24 and 48 hours after the cycling test, indicating that cherry juice consumption did not enhance recovery from exercise-induced muscle damage, compared with sports drink consumption. This might be due to the level of muscle damage induced by exercise. The cycling protocol in the current study most likely elicited a lower level of muscle damage compared to some of the protocols from previous studies. High-force eccentric contractions used in previous studies, such as resistance training or running, would probably elicit greater muscle damage than long-duration cycling, which does not involve an eccentric component. Tart cherry juice may be protective against larger amounts of muscle damage, but not smaller amounts elicited by cycling.

### Effects of tart cherry juice supplementation on blood pressure

No difference in post-exercise blood pressure was found between the two conditions, whereas many studies have found blood pressure-lowering effects of tart cherry juice [36–39]. One possible explanation is that it might take longer to detect blood pressure changes since blood pressure is reduced after six weeks [36] or twelve weeks [37] of tart cherry juice consumption.

Another factor could be participant type. Positive results were found in older adults between the ages of 65–80 years [37], individuals with type 2 diabetes [36], healthy adults with moderately elevated blood pressure [38], and individuals with early hypertension [39], whereas in the present study, we used young to middle-aged recreational cyclists (35±16 years old) who were in good health. Another study employing normotensive participants ($\sim$ 111/70 mm Hg) did not find positive results either [40]. Finally, timing of measurement might be an important factor. Significant reductions in systolic and diastolic blood pressure have been found 1.5 to 2 hours after tart cherry juice consumption [41], with return to baseline shortly thereafter [41]. In the present study, blood pressure was measured ~3.5–4 hours after tart cherry juice supplementation when the blood pressure might almost return to baseline levels. Blood pressure in the current study was measured post-exercise. The well-known post-exercise hypotension, which can last up to 24 hours [42], may have masked any effects of the tart cherry juice.

## Conclusion

Low-glycemic index tart cherry juice was not effective for improving cycling performance, carbohydrate and fat metabolism during exercise, and recovery from exercise in recreational cyclists, compared to consuming a high-glycemic index sports drink. Future studies are warranted for further elucidating the effects of tart cherry juice on muscle damage induced by exercise of greater intensity, especially with eccentric muscle contractions to compare recreationally active individuals and elite athletes. It should also be noted that although foods such as tart cherries may protect against muscle damage, the high amount of antioxidants may hinder long-term adaptation to exercise training (e.g., mitochondrial biogenesis) because much of the cell-signaling required for chronic adaptation involves signaling initiated by free-radicals produced during high rates of oxidative stress [43]. To see how tart cherry juice affects blood pressure acutely, the timing of the blood pressure measurement should be expanded to 48 hours post consumption. Finally, testing post-consumption insulin and glucose responses is necessary to investigate the mechanisms of how tart cherry juice affects substrate metabolism.

## Author Contributions

**Conceptualization:** Philip D. Chilibeck.

**Data curation:** Ruirui Gao, Nicole Rapin, Justin W. Andrushko, Jonathan P. Farthing, Julianne Gordon.

**Formal analysis:** Ruirui Gao.

**Funding acquisition:** Philip D. Chilibeck.

**Investigation:** Ruirui Gao, Nicole Rapin, Justin W. Andrushko, Julianne Gordon.

**Methodology:** Jonathan P. Farthing, Philip D. Chilibeck.

**Project administration:** Ruirui Gao, Philip D. Chilibeck.

**Resources:** Philip D. Chilibeck.

**Supervision:** Jonathan P. Farthing, Philip D. Chilibeck.

**Writing – original draft:** Ruirui Gao.

**Writing – review & editing:** Ruirui Gao, Justin W. Andrushko, Jonathan P. Farthing, Julianne Gordon, Philip D. Chilibeck.

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
