## [Decision Letter · Decision Letter 0]

24 Jan 2024

PONE-D-23-35326The effect of tart cherry juice compared to a sports drink on cycling exercise performance, substrate metabolism, and recoveryPLOS ONE

Dear Dr. Chilibeck,

Thank you for submitting your manuscript to PLOS ONE. After careful consideration, we feel that it has merit but does not fully meet PLOS ONE’s publication criteria as it currently stands. Therefore, we invite you to submit a revised version of the manuscript that addresses the points raised during the review process.

**Please consider all review comments to improve the readability of the text and provide enough experimental details.**

We look forward to receiving your revised manuscript.

Kind regards,

Franck Carbonero, PhD

Academic Editor

PLOS ONE

Journal Requirements:

This study was funded by the Canadian Cherry Producers and Saskatchewan Academy of Sports Medicine Inc.

This study was funded by the Canadian Cherry Producers and Saskatchewan Academy of Sports Medicine Inc.

This study was funded by the Canadian Cherry Producers and Saskatchewan Academy of Sports Medicine Inc.

4. In the online submission form, you indicated that The data underlying the results presented in the study are available from the corresponding author, Dr. Chilibeck (E-mail: phil.chilibeck@usask.ca).

Reviewers' comments:

Reviewer's Responses to Questions

**Comments to the Author**

1. Is the manuscript technically sound, and do the data support the conclusions?

Reviewer #1: Yes

Reviewer #2: Yes

2. Has the statistical analysis been performed appropriately and rigorously? 

Reviewer #1: Yes

Reviewer #2: Yes

3. Have the authors made all data underlying the findings in their manuscript fully available?

Reviewer #1: Yes

Reviewer #2: Yes

4. Is the manuscript presented in an intelligible fashion and written in standard English?

Reviewer #1: Yes

Reviewer #2: Yes

5. Review Comments to the Author

Reviewer #1: I appreciate the opportunity to read and review this manuscript. The authors provide a compelling study showing that tart cherry juice consumption as a low glycemic index (GI) drink compared with a sports drink (high GI) has no effect on performance, substrate oxidation, blood pressure and various recovery variables.

Major comments

• Given that non-trained athletes are not oxygen limited in terms of vigorous intensity aerobic exercise and performance on time-to exhaustion tests why would it be hypothesized that increased blood flow (more oxygen delivery) from increased vasodilation from tart cherry juice would enhance performance?

• Why would you expect muscle soreness or DOMS from this protocol? The intervention of 90min cycling at 65% VO2peak and a 10km time trial, that would primarily consist of concentric muscle contractions was used. It would not be expected for these participants to have DOMS or muscle damage since the protocol is not designed to cause muscle damage. All the muscle soreness articles referenced include eccentric damaging protocols.

• How is muscle soreness measured? The reference provided described a DOMS visual analog scale assessment, EMG with twitch, ultrasound of muscle thickness. It is not clear what methods you used. The units you provide in figure 2 for muscle soreness is in mm. What is that in reference to?

• The results sections should be longer and include all the outcome variables described and recorded.

• The discussion paragraph (lines 310-319) needs to include references because there are some specific statements made about glycemic index effecting lactate levels and how fructose is converted into lactate and this needs to have very specific references. I would also provide a reference and some detail from George A Brooks regarding lactate metabolism.

• Need to clarify the VO2 decreasing over exercise when the protocol was set at 65% of VO2 max with a fixed workload of 50% max. I would expect with this group of participants, that it would be more likely to see a VO2 slow component (a slow rise in VO2 during constant work rate exercise) as they might be working close to or above their lactate threshold. Was their lactate threshold determined on the preliminary VO2 max testing day?

Minor comments

• References should be included following the introduction statement. Also reference 1 is more focused on dietary nitrates, l-arginine and l-citrulline supplements and how they can improve performance. I don’t see how this relates to your intro on low glycemic tart cherry juice given they are rich in anthocyanins and flavonoids.

• How were anthocyanins measured in the tart cherry juice provided?

• Lines 116-117. You state participants were blinded to elapsed time then the following sentence it says participants were allowed to see elapsed time. Was the second sentence for the 90 min exercise condition?

• I don’t see any discussion of tart cherry (anthocyanins, flavonoids) or glycemic index on performance in reference 1.

• It would be good to introduce post exercise hypotension since your measure blood pressure as a secondary outcome. I would also include the data in the results on the 24-hr blood pressure monitor.

• You specifically mention in your conclusion that blood pressure measurements should be expanded from 1 hour to 48hr post. In the methods you describe that the participants wore a 24hr ambulatory blood pressure monitor after exercise and recorded every 30 minutes during waking hours and every 2 hours during sleep.

• It may be interesting to look at muscle recovery with tart cherry juice utilizing an eccentric muscle damaging protocol in both recreationally active individuals and elite athletes. You just mention in your conclusion that elite athletes would warrant further studies.

• Did any of the participants report muscle soreness?

• Was there a rest period between the 90 minute exercise and time trial?

• The figures are very low resolution.

Reviewer #2: Abstract

Page 2, lines 24-25: I understand that there is a word limit but consider rewriting the first sentence. It sounds awkward. Perhaps add “and” after the word properties.

“Tart cherries have low glycemic index, antioxidant and anti-inflammatory properties, and therefore, may benefit performance and enhance recovery from exercise.”

Page 2, line 28: Why is it stated that there were 12 recreational cyclists in the study but in parenthesis it provides the background data for 8 males? Is the background data just for the males or also including the females?

Introduction

Page 3, lines 46-47: Could you provide citations for studies that have shown tart cherry juice can inhibit inflammation and enhance recovery?

Page 3, lines 54-58: Can you add citations for the statement about tart cherries protective effect. Likewise, can you add citations for studies that have assessed muscle damage using indirect markers of muscle damage? Also could you address the literature that large doses of antioxidants, found in tart cherries, may also blunt exercise performance/improvements.

Page 3, lines 63-65: Why not include a placebo group as a third arm to the study?

Methods

Page 7, lines 140-142: Were participants instructed not to change their physical activity levels throughout the duration of the entire study, since they may be able to improve performance measures by increasing their physical activity and the amount of time they spend cycling?

Page 10-11, lines, 226-229: Could you write out/add the levels of the two-factor repeated measure ANOVA, such as 2(condition: tart cherry vs. high glycemic index drink) x 3( time: pre vs. immediately post vs. 24 h post vs. 48 h post)? Looking at the figures, it appears that the time levels were different for certain variables. This would make it more clear which variables were analyzed during what time.

Page 10-11, line 226-229: Was there a specific reason why the study was not designed to examine potential sex differences were not examined?

Page 11, lines 233-234: Could the Bonferroni corrected p-values be included in the results as well as the methods were applicable.

Results

Page 11-12, lines 245-254: I would suggest rewriting the results section to make it more clear for the readers. It is confusing the way it is currently. I would suggest breaking into more sentences. For example, all the dependent t-test results as one or a couple of sentences and then the repeated measure results as separate sentences. It is also unclear what is meant by “increasing over exercise” or “decreasing over exercise”. Does it mean a particular increased/decreased from each time point that was measured or only from one time to another (not all that were measured)? Lastly, please include the Bonferroni correct p-values where applicable.

Discussion

Page 14, lines 310-311: Please add citations for studies that investigated the effects of foods differing in glycemic index on lactate levels.

Page 15, lines 331-332: Please add citations for studies that have found blood pressure-lowering effects of tart cherry juice.

6. PLOS authors have the option to publish the peer review history of their article (what does this mean?). If published, this will include your full peer review and any attached files.

Reviewer #1: **Yes: **Catherine L. Jarrett

Reviewer #2: No

---

## [Author Response · Author response to Decision Letter 0]

2 Apr 2024

Response to editors:

We have removed mention of funding sources from the manuscript.

We have uploaded a file with all the study data

Response to reviewers

Thanks to the reviewers for taking the time to review our manuscript. We have responded to all comments below. Please note that references to page numbers are for the tracked changes version of the manuscript, which follows the clean, revised manuscript in the pdf. We refer to page numbers for revisions rather than line numbers because the line numbers shift when the journal converts our Word document to pdf.

Reviewer 1

Major comments

• Given that non-trained athletes are not oxygen limited in terms of vigorous intensity aerobic exercise and performance on time-to exhaustion tests why would it be hypothesized that increased blood flow (more oxygen delivery) from increased vasodilation from tart cherry juice would enhance performance?

Response: We have added the possibility that non-trained athletes might not be oxygen-limited during the testing in our study to the start of the discussion section: “Our study indicates that perhaps the participants in our study were not limited by oxygen delivery because they were not trained athletes and therefore any vasodilatory effect of the cherry juice would not have an effect on performance.” (page 14)

• Why would you expect muscle soreness or DOMS from this protocol? The intervention of 90min cycling at 65% VO2peak and a 10km time trial, that would primarily consist of concentric muscle contractions was used. It would not be expected for these participants to have DOMS or muscle damage since the protocol is not designed to cause muscle damage. All the muscle soreness articles referenced include eccentric damaging protocols.

Response: We agree with the reviewer that the optimal muscle-damaging protocol involves eccentric contractions; however, we would argue that high-intensity cycling can still result in muscle damage. For example, there is a growing number of cases of exertional rhabdomyolysis in people undertaking “spin” classes, cycling classes that would involve little eccentric contractions. There is also evidence for an elevation in markers of muscle damage with cycling. We have added this to the introduction, citing the references below.

Masuda Y, Wam R, Paik B, Ngoh C, Choong AM, Ng JJ. Clinical characteristics and outcomes of exertional rhabdomyolysis after indoor spinning: a systematic review. Phys Sportsmed. 2023 Aug;51(4):294-305. doi: 10.1080/00913847.2022.2049645. Epub 2022 Mar 10. PMID: 35254210.

Hebisz R, Borkowski J, Hebisz P. Creatine Kinase and Myoglobin Plasma Levels in Mountain Bike and Road Cyclists 1 h after the Race. Int J Environ Res Public Health. 2022 Aug 2;19(15):9456. doi: 10.3390/ijerph19159456. PMID: 35954814; PMCID: PMC9367889.

We have added the following to our introduction: “Although eccentric contractions are typically used in studies to induce muscle damage, high-intensity concentric contractions induced by repetitive cycling still has the potential to induce muscle damage.” (page 3)

We also highlight in our discussion the limitation that if we used a protocol involving eccentric contractions, we may have seen a greater amount of muscle damage.

• How is muscle soreness measured? The reference provided described a DOMS visual analog scale assessment, EMG with twitch, ultrasound of muscle thickness. It is not clear what methods you used. The units you provide in figure 2 for muscle soreness is in mm. What is that in reference to?

Response: Our apologies...somehow these methods were omitted from the manuscript. We have added greater details to the methods: 

“Using an Algometer (WAGNER Instruments, FDK 60 Force Dial; Greenwich, CT), 5kg of pressure was applied on four different sites of the knee extensors. While the participants were seated upright with a 90-degree bend in the knee, the distance between the anterior superior iliac crest and proximal edge of the patella was measured. Four locations were marked with a permanent marker: 25, 50, 75 and 100% of the distance between the two anatomical landmarks with 0% indicating the proximal landmark (anterior superior iliac crest). Participants reported pain on a 100mm visual analog scale, with 0mm indicating no pain and 100 mm indicating maximal pain. A total pain score was derived by taking an average of the pain from all four sites.” (page 6)

• The results sections should be longer and include all the outcome variables described and recorded.

Response: We double-checked the results section to ensure that all outcome variables were described. The only outcome variable we did not describe in full detail were the blood pressure results. These have been added to a table in the results section. We have expanded the results section to include all the post-hoc testing on the time main effects across different variables.

• The discussion paragraph (lines 310-319) needs to include references because there are some specific statements made about glycemic index effecting lactate levels and how fructose is converted into lactate and this needs to have very specific references. I would also provide a reference and some detail from George A Brooks regarding lactate metabolism.

Response: We have added references to support glycemic index affecting lactate levels and how fructose is converted to lactate in the liver, including a reference from George A Brooks:

Kaviani M, Chilibeck PD, Jochim J, Gordon J, Zello GA. The Glycemic Index of Sport Nutrition Bars Affects Performance and Metabolism During Cycling and Next-Day Recovery. J Hum Kinet. 2019 Mar 27;66:69-79. doi: 10.2478/hukin-2018-0050. PMID: 30988841; PMCID: PMC6458587.

Burns SP, Murphy HC, Iles RA, Bailey RA, Cohen RD. Hepatic intralobular mapping of fructose metabolism in the rat liver. Biochem J. 2000 Jul 15;349(Pt 2):539-45. doi: 10.1042/0264-6021:3490539. PMID: 10880353; PMCID: PMC1221177.

Brooks GA. What the Lactate Shuttle Means for Sports Nutrition. Nutrients. 2023 May 3;15(9):2178. doi: 10.3390/nu15092178. PMID: 37432330; PMCID: PMC10180760.

• Need to clarify the VO2 decreasing over exercise when the protocol was set at 65% of VO2 max with a fixed workload of 50% max. I would expect with this group of participants, that it would be more likely to see a VO2 slow component (a slow rise in VO2 during constant work rate exercise) as they might be working close to or above their lactate threshold. Was their lactate threshold determined on the preliminary VO2 max testing day?

Response: The lactate threshold was not determined; however, based on the lactate response to the exercise test (which decreased over the 90 minutes of cycling) we assume they were below their lactate threshold. The VO2 slow component correlates to blood lactate concentrations, so perhaps the increased removal of lactate during the test (i.e., through the Cori Cycle or lactate shuttle) may be associated with the slight reduction in VO2 during the test. 

We have added the following text to the discussion section: “There was a slight, but statistically significant decrease in VO2 across our exercise test (across both drink conditions) (Figure 1). This is contrary to the VO2 drift (i.e., slow component) that might be expected. The VO2 slow component is correlated with lactate concentration [32]. Lactate concentration decreased throughout our exercise test indicating our participants were most likely working below their lactate threshold. We speculate that the reduced VO2 observed during our exercise test may be related to the reduced lactate concentration, which is most likely due to the lactate shuttle or Cori Cycle during exercise [33].” (page 16)

Costa MM, Russo AK, Pićarro IC, Barros Neto TL, Silva AC, Tarasantchi J. Oxygen consumption and ventilation during constant-load exercise in runners and cyclists. J Sports Med Phys Fitness. 1989 Mar;29(1):36-44. PMID: 2770266.

Brooks GA. What the Lactate Shuttle Means for Sports Nutrition. Nutrients. 2023 May 3;15(9):2178. doi: 10.3390/nu15092178. PMID: 37432330; PMCID: PMC10180760.

Minor comments

• References should be included following the introduction statement. Also reference 1 is more focused on dietary nitrates, l-arginine and l-citrulline supplements and how they can improve performance. I don’t see how this relates to your intro on low glycemic tart cherry juice given they are rich in anthocyanins and flavonoids.

Response: We have added the following references to support the opening sentence on the potential for tart cherry juice to improve recovery from exercise and exercise performance. We also replaced reference 1 with a reference that is focused on tart cherry juice (the first reference listed below).

Connolly DA, McHugh MP, Padilla-Zakour OI, Carlson L, Sayers SP. Efficacy of a tart cherry juice blend in preventing the symptoms of muscle damage. Br J Sports Med. 2006 Aug;40(8):679-83; discussion 683. doi: 10.1136/bjsm.2005.025429. Epub 2006 Jun 21. PMID: 16790484; PMCID: PMC2579450.

Bowtell JL, Sumners DP, Dyer A, Fox P, Mileva KN. Montmorency cherry juice reduces muscle damage caused by intensive strength exercise. Med Sci Sports Exerc. 2011 Aug;43(8):1544-51. doi: 10.1249/MSS.0b013e31820e5adc. PMID: 21233776.

Gao R, Chilibeck PD. Effect of Tart Cherry Concentrate on Endurance Exercise Performance: A Meta-analysis. J Am Coll Nutr. 2020 Sep-Oct;39(7):657-664. doi: 10.1080/07315724.2020.1713246. Epub 2020 Jan 27. PMID: 31986108.

• How were anthocyanins measured in the tart cherry juice provided?

Response: Anthocyanins were estimated based on the Cherry Juice product label and not measured directly. This has been added as a limitation in the discussion section. (page 16)

• Lines 116-117. You state participants were blinded to elapsed time then the following sentence it says participants were allowed to see elapsed time. Was the second sentence for the 90 min exercise condition?

Response: Participants were blinded to elapsed time during the 10km time trial. The following sentence indicates that they were allowed to see the elapsed distance. (page 6)

• I don’t see any discussion of tart cherry (anthocyanins, flavonoids) or glycemic index on performance in reference 1.

Response: This reference has been replaced with a few references that mention the effect of tart cherries on recovery from exercise and performance.

• It would be good to introduce post exercise hypotension since your measure blood pressure as a secondary outcome. I would also include the data in the results on the 24-hr blood pressure monitor.

Response: We add the following to the discussion section: “Blood pressure in the current study was measured post-exercise. The well-known post-exercise hypotension, which can last up to 24 hours, may have masked any effects of the tart cherry juice.” (page 18)

Alpsoy Ş. Exercise and Hypertension. Adv Exp Med Biol. 2020;1228:153-167. doi: 10.1007/978-981-15-1792-1_10. PMID: 32342456.

We have also added a table in the results section describing in more detail the 24-hour blood pressure results.

• You specifically mention in your conclusion that blood pressure measurements should be expanded from 1 hour to 48hr post. In the methods you describe that the participants wore a 24hr ambulatory blood pressure monitor after exercise and recorded every 30 minutes during waking hours and every 2 hours during sleep.

Response: This is an error in our conclusion. We have deleted mention of the 1 hour measurement to simply state that measurements could be expanded to 48 hours.

• It may be interesting to look at muscle recovery with tart cherry juice utilizing an eccentric muscle damaging protocol in both recreationally active individuals and elite athletes. You just mention in your conclusion that elite athletes would warrant further studies.

Response: We have modified our conclusion to suggest this as a future direction.

• Did any of the participants report muscle soreness?

Response: Muscle soreness results are presented in bottom panel of figure 2. Overall, the increase in muscle soreness was not statistically significant (p=0.078), but given this trend, some of the participants would have reported muscle soreness, especially immediately post-exercise.

• Was there a rest period between the 90 minute exercise and time trial?

Response: There was no rest between the 90 minute exercise and the time trial. This is clarified in the methods where we now state the time trial was performed immediately after the 90-minutes of exercise.

• The figures are very low resolution.

Response: We have submitted high-resolution figures as individual figure files (tifs). The resolution may have been lost when the manuscript was converted to pdf for review.

Reviewer #2: Abstract

Page 2, lines 24-25: I understand that there is a word limit but consider rewriting the first sentence. It sounds awkward. Perhaps add “and” after the word properties.

“Tart cherries have low glycemic index, antioxidant and anti-inflammatory properties, and therefore, may benefit performance and enhance recovery from exercise.”

Response: This has been corrected as suggested

Page 2, line 28: Why is it stated that there were 12 recreational cyclists in the study but in parenthesis it provides the background data for 8 males? Is the background data just for the males or also including the females?

Response: When we put in “8 males” we assumed the reader would be able to determine the remaining participants were females and that the background data referred to all 12 participants. To make this clearer we have added “4 females” in the parenthesis.

Introduction

Page 3, lines 46-47: Could you provide citations for studies that have shown tart cherry juice can inhibit inflammation and enhance recovery?

Response: We have added several references here, as suggested:

Connolly DA, McHugh MP, Padilla-Zakour OI, Carlson L, Sayers SP. Efficacy of a tart cherry juice blend in preventing the symptoms of muscle damage. Br J Sports Med. 2006 Aug;40(8):679-83; discussion 683. doi: 10.1136/bjsm.2005.025429. Epub 2006 Jun 21. PMID: 16790484; PMCID: PMC2579450.

Bowtell JL, Sumners DP, Dyer A, Fox P, Mileva KN. Montmorency cherry juice reduces muscle damage caused by intensive strength exercise. Med Sci Sports Exerc. 2011 Aug;43(8):1544-51. doi: 10.1249/MSS.0b013e31820e5adc. PMID: 21233776.

Gao R, Chilibeck PD. Effect of Tart Cherry Concentrate on Endurance Exercise Performance: A Meta-analysis. J Am Coll Nutr. 2020 Sep-Oct;39(7):657-664. doi: 10.1080/07315724.2020.1713246. Epub 2020 Jan 27. PMID: 31986108.

Page 3, lines 54-58: Can you add citations for the statement about tart cherries protective effect. Likewise, can you add citations for studies that have assessed muscle damage using indirect markers of muscle damage? Also could you address the literature that large doses of antioxidants, found in tart cherries, may also blunt exercise performance/improvements.

Response: We have again cited the following studies to support the statement on cherries protective effect using indirect markers of muscle damage:

Connolly DA, McHugh MP, Padilla-Zakour OI, Carlson L, Sayers SP. Efficacy of a tart cherry juice blend in preventing the symptoms of muscle damage. Br J Sports Med. 2006 Aug;40(8):679-83; discussion 683. doi: 10.1136/bjsm.2005.025429. Epub 2006 Jun 21. PMID: 16790484; PMCID: PMC2579450.

Bowtell JL, Sumners DP, Dyer A, Fox P, Mileva KN. Montmorency cherry juice reduces muscle damage caused by intensive strength exercise. Med Sci Sports Exerc. 2011 Aug;43(8):1544-51. doi: 10.1249/MSS.0b013e31820e5adc. PMID: 21233776.

Regarding the comment that excessive antioxidant supplementation may blunt exercise performance: I think this is more of a chronic effect; however, to address your point, we have added a statement on this in the conclusion of the manuscript: 

“It should also be noted that although foods such as tart cherries may protect against muscle damage, the high amount of antioxidants may hinder long-term adaptation to exercise training (e.g., mitochondrial biogenesis) because much of the cell-signaling required for chronic adaptation involves signaling initiated by free-radicals produced during h

---

## [Decision Letter · Decision Letter 1]

30 Apr 2024

PONE-D-23-35326R1The effect of tart cherry juice compared to a sports drink on cycling exercise performance, substrate metabolism, and recoveryPLOS ONE

Dear Dr. Chilibeck,

Thank you for submitting your manuscript to PLOS ONE. After careful consideration, we feel that it has merit but does not fully meet PLOS ONE’s publication criteria as it currently stands. Therefore, we invite you to submit a revised version of the manuscript that addresses the points raised during the review process.

Please answer Reviewer 1's questions and implement their minor edits suggestions.

We look forward to receiving your revised manuscript.

Kind regards,

Franck Carbonero, PhD

Academic Editor

PLOS ONE

Journal Requirements:

Additional Editor Comments:

Please answer Reviewer 1's questions and implement their minor edits suggestions.

Reviewers' comments:

Reviewer's Responses to Questions

**Comments to the Author**

1. If the authors have adequately addressed your comments raised in a previous round of review and you feel that this manuscript is now acceptable for publication, you may indicate that here to bypass the “Comments to the Author” section, enter your conflict of interest statement in the “Confidential to Editor” section, and submit your "Accept" recommendation.

Reviewer #1: All comments have been addressed

Reviewer #2: All comments have been addressed

2. Is the manuscript technically sound, and do the data support the conclusions?

Reviewer #1: Yes

Reviewer #2: Yes

3. Has the statistical analysis been performed appropriately and rigorously? 

Reviewer #1: Yes

Reviewer #2: Yes

4. Have the authors made all data underlying the findings in their manuscript fully available?

Reviewer #1: Yes

Reviewer #2: Yes

5. Is the manuscript presented in an intelligible fashion and written in standard English?

Reviewer #1: Yes

Reviewer #2: Yes

6. Review Comments to the Author

Reviewer #1: The authors provided detailed responses and edited the manuscript with more references regarding tart cherry juice. They also clarified methods that were lacking and data that was missing from the results. I selected minor revisions, because I would like the authors to confirm that power output was maintained during the 90 minutes. I think the results would benefit from a figure that shows the power output during the 90min cycling test. One minor comment is the first panel of figure one y-axis is labeled "pre-exercise glucose" and the x axis is labeled with times post exercise. Also, it doesn't appear that post exercise blood pressure is discussed in the results section. The authors added results regarding excluded participant data that was not in the original version, the participant numbers should be included in the figure/table legends.

Reviewer #2: Thank you to the authors for responding to my comments. The manuscript is well written and well thought out! Great work!

7. PLOS authors have the option to publish the peer review history of their article (what does this mean?). If published, this will include your full peer review and any attached files.

Reviewer #1: No

Reviewer #2: No

---

## [Author Response · Author response to Decision Letter 1]

28 Jun 2024

Reviewer #1: The authors provided detailed responses and edited the manuscript with more references regarding tart cherry juice. They also clarified methods that were lacking and data that was missing from the results. I selected minor revisions, because I would like the authors to confirm that power output was maintained during the 90 minutes. I think the results would benefit from a figure that shows the power output during the 90min cycling test. 

Response: Power output was maintained throughout the test by the researcher watching closely that the participants maintained the required pedaling rate. We have added a statement on this to the manuscript (lines 164-166). We do not have the data for a figure showing the power output during the test. 

One minor comment is the first panel of figure one y-axis is labeled "pre-exercise glucose" and the x axis is labeled with times post exercise. 

Response: The “post” indicators on the x-axis of the graph refer to post-drink consumption and not post-exercise. We have clarified this in the legend for the figure (line 270) .

Also, it doesn't appear that post exercise blood pressure is discussed in the results section. 

Response: The blood pressure results are mentioned on line in the results section (line 275).

The authors added results regarding excluded participant data that was not in the original version, the participant numbers should be included in the figure/table legends.

Response: We have added the participant numbers to the figure legends and blood pressure table, and in the text of the results section as suggested.

---

## [Editor Report · Decision Letter 2]

2 Jul 2024

The effect of tart cherry juice compared to a sports drink on cycling exercise performance, substrate metabolism, and recovery

PONE-D-23-35326R2

Dear Dr. Chilibeck,

We’re pleased to inform you that your manuscript has been judged scientifically suitable for publication and will be formally accepted for publication once it meets all outstanding technical requirements.

Kind regards,

Franck Carbonero, PhD

Academic Editor

PLOS ONE
---

## [Editor Report · Acceptance letter]

4 Jul 2024

PONE-D-23-35326R2 

PLOS ONE

Dear Dr. Chilibeck, 

I'm pleased to inform you that your manuscript has been deemed suitable for publication in PLOS ONE. Congratulations! Your manuscript is now being handed over to our production team.

Kind regards, 

on behalf of

Dr. Franck Carbonero 

Academic Editor

PLOS ONE